# An Indoor UWB 3D Positioning Method for Coplanar Base Stations

**DOI:** 10.3390/s22249634

**Published:** 2022-12-08

**Authors:** Ning Zhou, Minghao Si, Dehai Li, Chee Kiat Seow, Jinzhong Mi

**Affiliations:** 1Chinese Academy of Surveying and Mapping, Beijing 100036, China; 2School of Environmental Science and Spatial Informatics, China University of Mining and Technology, Xuzhou 221116, China; 3School of Computing Science, University of Glasgow, Sir Alwyn Williams Building, Glasgow G12 8RZ, UK

**Keywords:** UWB, indoor location, iteration algorithm, coplanar base station

## Abstract

As an indispensable type of information, location data are used in various industries. Ultrawideband (UWB) technology has been used for indoor location estimation due to its excellent ranging performance. However, the accuracy of the location estimation results is heavily affected by the deployment of base stations; in particular, the base station deployment space is limited in certain scenarios. In underground mines, base stations must be placed on the roof to ensure signal coverage, which is almost coplanar in nature. Existing indoor positioning solutions suffer from both difficulties in the correct convergence of results and poor positioning accuracy under coplanar base-station conditions. To correctly estimate position in coplanar base-station scenarios, this paper proposes a novel iterative method. Based on the Newton iteration method, a selection range for the initial value and iterative convergence control conditions were derived to improve the convergence performance of the algorithm. In this paper, we mathematically analyze the impact of the localization solution for coplanar base stations and derive the expression for the localization accuracy performance. The proposed method demonstrated a positioning accuracy of 5 cm in the experimental campaign for the comparative analysis, with the multi-epoch observation results being stable within 10 cm. Furthermore, it was found that, when base stations are coplanar, the test point accuracy can be improved by an average of 63.54% compared to the conventional positioning algorithm. In the base-station coplanar deployment scenario, the upper bound of the CDF convergence in the proposed method outperformed the conventional positioning algorithm by about 30%.

## 1. Introduction

In recent years, positioning services have become necessary in daily life. While, GNSS can provide accurate three-dimensional (3D) positions in outdoor environments, UWB positioning is considered a reliable positioning method in indoor environments due to its high ranging accuracy. With the development of two-way ranging (TWR) technology, UWB positioning can achieve centimeter-scale ranging with unsynchronized clocks between stations [1,2,3]. Many researchers have built indoor positioning systems based on UWB technology, which can achieve centimeter-level positioning accuracy. These researchers analyzed the usability of UWB technology and showed its effectiveness with a TWR ranging technique in harsh environments [4,5].

However, there are still some problems in UWB positioning systems. Compared with GNSS, which can provide accurate elevation, there are various limitations in the elevation calculations of UWB system. In the outdoor environment, GNSS satellite positions are all located in the zenith direction, and the elevation accuracy of the unknown point solution results is worse than the plane positioning accuracy. Different GNSS satellites have different orbital altitudes, and even satellites at the same orbital height have differences in altitude angles, which helps to avoid extreme situations in which the same satellite height is used for the unknown point solution. In downhole positioning environments, the base-station deployment height is limited to a narrow interval near the ceiling, and the unknown point positioning solution lacks control in the elevation direction, making it difficult to obtain high-precision positioning results. Therefore, the precision of the z-axis coordinates is worse than that of the x-axis and y-axis coordinates. In outdoor environments, the surface heights do not overlap, and one horizontal position corresponds to one elevation value. High-precision elevation can be obtained by extrapolating the information from the Earth’s gravity field. However, with UWB positioning, the same horizontal positions for people, cars, and equipment are at different heights. In order to achieve control over equipment and personnel safety, a high-precision 3D position is essential.

However, available methods do not take into account the needs of coplanar base-station deployment and 3D positioning simultaneously. In this study, an existing iterative algorithm was improved for coplanar base-station 3D positioning. The method achieves the correct convergence of the results through two steps: initial value selection and iterative control. Furthermore, this paper provides theoretical support and analysis for 3D localization in coplanar base-station scenarios. The contributions include the following:
(1)Initial value selection scheme: For an indoor positioning scenario with coplanar deployment of base stations, we analyzed the influence of base-station deployment on the iterative calculation. We present a method for avoiding the computational difficulties caused by initial value selection;(2)Convergence control: Since there may be multiple extreme points in the nonconvex function, we applied a convergence control method to ensure convergence to the correct solution;(3)Theoretical analysis: The accuracy of the positioning results was theoretically calculated based on the least-squares method. The influence of base-station coplanar deployment on the calculation of the positioning equation was analyzed. In addition, the effect of a near-coplanar base-station equation on the results was derived.

After the improvement of the method, the localization solution could converge correctly in the case of coplanar base stations, and the localization accuracy was less than 10 cm, which provides the possibility of high-accuracy localization in cases of special base-station deployment.

The rest of the paper is organized as follows: Section 2 briefly describes the localization method based on the least-squares criterion. Section 3 proposes a new method adapted to the coplanar deployment of base stations. Section 4 describes simulations and the experimental campaign for the new algorithm. Section 5 discusses the impact of the base-station coplanar deployment on computation, and Section 6 concludes the paper and shows the limitations of the algorithm.

## 2. Related Work

### 2.1. Traditional Distance-Based Indoor Position Method

To facilitate the description of the indoor positioning model, a three-dimensional cartesian coordinate system was set, where the x- and y-axis are horizontal and the h-axis is vertical under the right-hand rule. The measured distance, unknown point, and known location for the base station have the following relationship: (1)$$d_{i}=\sqrt{{\left(P_{u}-P_{i}\right)}^{T}\left(P_{u}-P_{i}\right)}$$

The base station at the known location is marked as $P_{1}\left(x_{1},y_{1},h_{1}\right),\cdots P_{n}\left(x_{n},y_{n},h_{n}\right)$. The measured distance between the mobile station and each base station is represented by $d_{1},\cdots ,d_{n}$, and the location of the unknown mobile point is denoted $P_{u}\left(x,y,h\right)$. The system of equations is solved for the unknown point coordinates as follows: (2)$${\left(x-x_{1}\right)}^{2}+{\left(y-y_{1}\right)}^{2}+{\left(h-h_{1}\right)}^{2}=d_{1}^{2}\;\vdots \;{\left(x-x_{n}\right)}^{2}+{\left(y-y_{n}\right)}^{2}+{\left(h-h_{n}\right)}^{2}=d_{n}^{2}$$

The above relationship between the observed value and the coordinate value of the point to be found in Equation (2) can be transformed into an error equation, which is expressed as a matrix as $\;V=F\left(X\right)-D$. The vector ${\left[\begin{array}{ccc} v_{1} & \ldots & v_{n} \end{array}\right]}^{T}$ is denoted V, $X={\left[x^{2}+y^{2}+h^{2},x,y,z\right]}^{T}$, $F\left(X\right)$ is a function of X, and the constant term ${\left[d_{1}^{2}-x_{1}^{2}-y_{1}^{2}-h_{1}^{2},\ldots ,d_{n}^{2}-x_{n}^{2}-y_{n}^{2}-h_{n}^{2}\right]}^{T}$ is represented as  D; Rearranging Equation (2) gives the following: (3)$$\left[\begin{array}{c} v_{1}\\ \vdots \\ v_{2} \end{array}\right]=\left[\begin{array}{cccc} 1 & -2x_{1} & -2y_{1} & -2h_{1}\\ \vdots & \vdots & \vdots & \vdots \\ 1 & -2x_{n} & -2y_{n} & -2h_{n} \end{array}\right]\left[\begin{array}{c} x^{2}+y^{2}+h^{2}\\ x\\ y\\ h \end{array}\right]-\left[\begin{array}{c} d_{1}^{2}-x_{1}^{2}-y_{1}^{2}-h_{1}^{2}\\ \vdots \\ d_{n}^{2}-x_{n}^{2}-y_{n}^{2}-h_{n}^{2} \end{array}\right]$$
where the solution results obey the following constraint limits: (4)$$X\in \left\{{\left(x_{0},x_{1},x_{2},x_{3}\right)}^{T}\in R^{𝟜}|x_{0}=x_{1}^{2}+x_{2}^{2}+x_{3}^{2}\right\}$$

Direct methods and iterative methods can both be used in solving the above equations.

According to the least-squares criterion, the direct method solves the *X* expression as: (5)$$X=\mathrm{argmin}||F\left(X\right)-D{||}^{2}$$

After linearizing and organizing Equation (5), B denotes the X coefficient array and L denotes the constant-term array, with the superscript ^ indicating the estimated value. $\hat{X}$ can be expressed as: (6)$$\hat{X}={\left(B^{T}B\right)}^{-1}B^{T}L$$

Iterative methods are widely used for the solution of nonlinear least-squares problems [6]. Under the minimum sum of squares of errors criterion, the optimized method can be applied to obtain the solutions for the unknown points for indoor positioning. The objective function is convex and quadratically differentiable in the solution interval. Newton’s method can be used to iterate continuously toward the minimum residual sum of squares and find the target position, and the solution pseudocode is as follows.
**Algorithm** The Newton’s method of solving the position algorithmStep = 0; Iteration initial value = $X_{0}$; $g_{k}=F^{\prime }\left(X_{0}\right)$, $H_{0}=\partial_{xx^{T}}^{2}F\left(X_{0}\right)$; Maximum number of iterative calculations = k;*While* ($\left|X_{k+1}-X_{k}\right|<threshold$) and iterations < k $\;\;t_{k}Q\left(x_{k}\right)={\left[\partial_{xx^{T}}^{2}F\left(x_{k}\right)\right]}^{-1}$; $\;\;x_{k}=x_{k}-g_{k}H_{k}^{-1}$; $\;\;g_{k}=F^{\prime }\left(X_{k}\right)$; $\;\;H_{k}=\partial_{xx^{T}}^{2}F\left(x_{k}\right)$;   k=k+1; *End*
*Return*
$X_{k}$


More common is the TS-LS method. A set of nonlinear measurement equations is linearized and expanded in a Taylor series at a point that serves as the initial true position estimate. This set of linearized equations is solved to produce a new approximate position, and the process continues until a pre-specified criterion is satisfied [7]. Considered the TOA distance equation, the expression for the objective function matrix is as follows: (7)$$\left[\begin{array}{ccc} \frac{x-x_{1}}{{\hat{d}}_{1}} & \frac{y-y_{1}}{{\hat{d}}_{1}} & \frac{h-h_{1}}{{\hat{d}}_{1}}\\ \vdots & \vdots & \vdots \\ \frac{x-x_{n}}{{\hat{d}}_{n}} & \frac{y-y_{n}}{{\hat{d}}_{n}} & \frac{h-h_{n}}{{\hat{d}}_{n}} \end{array}\right]\left[\begin{array}{c} x\\ y\\ h \end{array}\right]=\left[\begin{array}{c} d_{1}-{\hat{d}}_{1}\\ \vdots \\ d_{n}-{\hat{d}}_{n} \end{array}\right]+\left[\begin{array}{c} v_{1}\\ \vdots \\ v_{n} \end{array}\right]$$

This can be written in a compact form as: (8)AX=L+V

The weighted LS solution of Equation (8) is: (9)$$X={\left(A^{T}A\right)}^{-1}\left(A^{T}\left(L+V\right)\right)$$

There are many iterative optimization methods available for the iterative solution process; for example, the Levenberg–Marquardt method [8], Broyden–Fletcher–Goldfarb–Shanno (BFGS) methods [9], and so on.

### 2.2. Advanced Research

In order to achieve 3D positioning in multiple scenes, many studies have been undertaken, including studies in specific positioning environments. Some researchers first narrowed the localization scope and found a possible solution interval [10]. In scenarios where 3D positioning needs to be acquired, researchers have improved the dispersion of the base station in the altitude direction [11,12,13,14]. One method is to deploy a dense grid of anchor networks [15]. The introduction of new sources of information is another type of method used to obtain 3D positioning. Some studies have used geometric relationships to estimate localization results [16,17]. The authors of [18] combined ranging characteristics with map features for localization. Another study [19] used RFID to localize people. Magneto-inductive technology has also been used in 3D positioning due to its ability to obtain angular information [20]. Boosting algorithms are another class of methods used to make results converge correctly [21,22,23]. In the least-squares iterative method-based calculation, the algorithm improves the positioning accuracy by obtaining accurate initial values [24], while other methods improve the convergence of the algorithms [16].

When it comes to coplanar localization of base stations, these methods lack adaptive treatment of the environment, making them less effective. New improvements are needed for the traditional methods, and the related work is presented in Section 3.

## 3. A Proposed Method for Base-Station Coplanar Iterations

### 3.1. Iterative Initial Value Selection

When using the iterative method, as algorithim1 shows, selecting an initial value close to the true value is beneficial for correct convergence. When dealing with convex functions, the poles are single, the iterations are always updated in the direction of the poles, and the choice of initial values does not affect the correctness of the final result. In the case of nonconvex functions, there may be multiple poles forming different first-order derivative descent regions, and the iterative solution may be near any one of these poles.

When the base station is laid out near the plane, an unknown point can cause the correct solution to be indistinguishable from the base-station high point if it is too close to the base-station layout plane. In this paper, we assume that the base station is deployed near the top surface of the space, all the unknown points are located below the base station, the base station is uniformly deployed in the x and y directions, and the mobile unknown points are located inside the base station envelope. $x_{ANCmin}<x_{TAG}<x_{ANCmax}$, $y_{ANCmin}<y_{TAG}<y_{ANCmax}$, and $h_{TAG}<h_{ANC}$. The subscript TAG indicates a mobile unknown point, and ANC indicates a known base station.

Newton’s method was chosen to iteratively calculate the value of the unknown point, and the method can converge correctly in the x and y directions by taking into account the calculation of the height of the near-plane deployment. The solution space for the mobile station location is: (10)X=xyh∈x,y,hT∈ℝ3xANCmin<x<xANCmaxyANCmin<y<yANCmaxh<hANCmin<hANCmax

The superscript ~ indicates the true value. There are two extreme-value points for the objective function in the solution space: (11)$$\left\{\begin{array}{c} X_{TAG1}\approx {\left[\tilde{x},\tilde{y},\tilde{h}\right]}^{T}\\ X_{TAG2}\approx {\left[\tilde{x},\tilde{y},h_{ANC}\right]}^{T} \end{array}\right.$$

Bounded by the extreme-value point near the true value, the initial value for the *h* direction is selected in the two intervals; if the iterative initial value is between the true value and the base station (i.e., $\tilde{h}<h_{0}<h_{ANC}$), the iteration process may tend toward any extreme-value point. If there is reliable prior localization information about the unknown point, the model may be able to control the iteration to avoid the incorrect extreme-value point, but this is often difficult to achieve in practice. If the initial value of the iteration is smaller than the true value, $h_{0}<\tilde{h}<h_{ANC}$, the iterative process will necessarily approach the minimum value near the true value first. Therefore, choosing initial values in the h direction that are smaller than the true value will facilitate iteration toward the correct solution.

It is assumed that the a priori information is the relative relationship between the unknown point and the base-station plane, the coordinates of the base station are known, and the observed value is the distance between the unknown point and each base station. The initial value $x_{o,}y_{o}$ of iteration in the x and y directions can be set as any value in the solution space, and the initial value $h_{o\;}\;$ of iteration in the h direction is set as the minimum value for the height in the solution space. If the height has a priori information, $h_{o\;}\;$ is set to any value less than the height of the unknown point; that is,
(12)$$X\in \left\{\left.{\left(x_{o},y_{o},h_{o}\right)}^{T}\in R^{𝟛}\right|\right.\left.x_{ANCmin}<x_{o}<x_{ANCmax},y_{ANCmin}<y_{o}<y_{ANCmax},h_{o}<\tilde{h_{TAG}}\right\}$$

In this case, using Newton’s method, the model will iterate in the h direction in an increasing manner; it will first determine the vicinity of the correct solution and can then be made to converge toward the correct solution in the elevation direction by choosing an appropriate step size.

### 3.2. Iterative Process Controls

The initial value selection process provides some help in the calculation but, in practice, the iteration points may cross the “valley” near the correct extremum due to the slow update in the x and y directions. If the iteration step is too large, the iteration point may jump out of the correct solution. Therefore, it is necessary to control the iteration step size based on the initial value selection.

The calculations for the iterative Newton’s method employed in the positioning algorithm are as follows: (13)$$X_{k+1}=X_{k}-{\left(\frac{\partial^{2}V^{T}V}{\partial X^{2}}\right)}^{-1}\left(\frac{\partial V^{T}V}{\partial X}\right)$$

The iteration step is 1, and the size of the iterative update value is determined by the first-order derivative and second-order derivative of the function. Near the extreme value, the value of the first-order derivative gradually decreases, and the inverse of the second-order derivative may be larger, resulting in the direction value being larger, the iteration update value changing too fast, and the convergence becoming problematic, causing the model to jump out of the iterative solution space in the positioning environment. Therefore, it is necessary to increase the step size to limit the iterations.

Assume that centimeter-level accuracy is required for the calculated positioning result, and the step adjustment coefficients are denoted $\mathrm{diag}\left(t_{x},t_{y},t_{h}\right)$. $t_{x}$, $t_{y}$, and $t_{h}$ denote the step adjustment parameters for the three directions and are set to $10^{-c}$, where C is a nonnegative constant. After the adjustment, all three directional components are updated to the centimeter level, which is represented as: (14)$$\left[\begin{array}{c} \Delta x\\ \Delta y\\ \Delta h \end{array}\right]=\left[\begin{array}{ccc} t_{x} & 0 & 0\\ 0 & t_{y} & 0\\ 0 & 0 & t_{h} \end{array}\right]\left[\begin{array}{ccc} \frac{\partial V^{T}V}{\partial x^{2}} & \frac{\partial V^{T}V}{\partial x\partial y} & \frac{\partial V^{T}V}{\partial x\partial h}\\ \frac{\partial V^{T}V}{\partial y\partial x} & \frac{\partial V^{T}V}{\partial y^{2}} & \frac{\partial V^{T}V}{\partial y\partial h}\\ \frac{\partial V^{T}V}{\partial h\partial x} & \frac{\partial V^{T}V}{\partial h\partial y} & \frac{\partial V^{T}V}{\partial h^{2}} \end{array}\right]\left[\begin{array}{c} \frac{\partial V^{T}V}{\partial x}\\ \frac{\partial V^{T}V}{\partial y}\\ \frac{\partial V^{T}V}{\partial h} \end{array}\right]<\left[\begin{array}{c} 0.1\\ 0.1\\ 0.1 \end{array}\right]$$

If the first-order bias derivative jumps positively or negatively, this means that the current step size cannot be used to approach the extreme-value point. The step adjustment value can be further reduced to iterate at the millimeter scale and improve the accuracy of the positioning results.

Since there are multiple first-order derivative zeros in the coplanar direction of the base station, the iterative calculation may still cross the correct solution after limiting the choice of the initial value. Therefore, restrictions are added to the first-order derivative iteration values during the generation process to avoid this situation.

After the initial value is selected, the first-order bias value of the h-axis increases continuously from the bottom of the measurement space and reaches the first zero point near the true value. The h value continues to increase, the first-order bias value rises and then falls, and the second zero point exists at the base station deployment height. In accordance with the variation, the iteration of h is restricted to the increasing direction during localization estimation. When the first-order partial derivative value at the new iteration point is smaller than the previous value, the continued iteration may converge to the wrong extreme-value point. In this case, the value of the iteration point H is reduced to ensure that the iteration is updated in the correct solution neighborhood, and the model iterates in the direction of the increasing first-order derivatives of H.

In general, after the initial value selection and iteration step restriction, the intermediate value of the iteration will not slip into the extreme point near the base-station deployment elevation. The strongly set constraint exists only as a guarantee of the calculation process.

The flow chart for the new algorithm with initial value selection and iterative process control is shown in Figure 1.

## 4. Experiment

### 4.1. Multitest Point-Positioning Simulation

In this section, we describe the simulations performed to test the proposed method, and then the TSLS and ILS were chosen as benchmarks. Coordinate accuracy and algorithm operation time were both considered to assess the performance of the proposed method. In addition, experiments were carried out for different distance observation errors. The simulation scenario is shown in Figure 2.

Base0station and test0point configuration: The coplanar station layout is shown in Figure 2. Four base stations, the coordinates of which were known, were set at the four corners of the ceiling in the test space, which means that the activity range of all test points was below the height of the base stations, and their planar coordinate activity range was limited. The coordinates were as follows: A1(1, 1, 3), A2(1, 13, 3), A3(13, 13, 3), and A4(13, 1, 3). Fifteen test points were randomly generated in the test space.

Distance observation: Observations were generated for distances between every base station and every test point, and random errors obeying $N\left(0,5^{2}\right)$ were added to simulate the reality, the unit of which was centimeters. All observations were assumed to be unaffected by occlusion and multipath points. One hundred epochs of observations were generated at each test point: $x=d_{1},d_{2},d_{3},d_{4}$.

#### 4.1.1. Internal Compliance Accuracy

The results calculated with three methods were compared to the observed values. The first was the restricted iterative least-squares (ILS) method introduced in Section 2, the second was the classical Taylor series least-squares (TSLS) solution method, and the third was the method proposed in this paper. The estimation results for multiple epochs showed that these methods differed in accuracy and stability, and the cumulative distribution function (CDF) reflects the magnitude of the localization result bias. The upper bound of the CDF convergence results for all estimations for each point was taken to represent the difference in localization accuracy between the three methods. The stability of the results was determined by calculating the variance in the estimates for each test point. Table 1 gives the specific statistical values for each of the 15 test points. It can be seen that the proposed method substantially outperformed the ILS [7] and TSLS methods in terms of accuracy and stability.

The statistics show that the estimation results obtained with the proposed method could all converge within 0.1 m, while most of the ILS results were outside of 1 m, and the TSLS results mostly varied within 2 m. The standard deviation (SD), as a reference indicator of internal compliance accuracy, indicated that the results of the proposed method were more concentrated than the ILS and TSLS results, which makes it easier to find inconsistent values and reject coarse differences during practical applications. This difference is visualized in Figure 3.

#### 4.1.2. External Compliance Accuracy

The internal conformity accuracy reflects the characteristics of the results themselves, and a comparison of the results with the true values can demonstrate the accuracy of the results. Both aspects illustrate the advantages and disadvantages of different method estimation results. A comparison of the estimated external conformity was undertaken. Each epoch’s results were averaged to obtain a final value for the point location estimation. The main difference in the positioning accuracy between the three methods was in the elevation direction. The results are shown in Table 2.

The comparison between the estimation results of the three methods is shown in Figure 4. The multi-epoch localization results of these methods differed significantly in accuracy and stability, where the red crosses indicate data outliers.

The difference between the upper and lower quartiles of the estimation error of the proposed method was stable within 10 cm, and the difference between the median and 0 did not exceed 5 cm. Excluding some outliers, the absolute values of most of the estimated absolute errors were less than 10 cm. This means that the average or median of the multi-epoch results from the observations should be closer to the true values. Extending the observation time or increasing the sampling frequency would improve the positioning accuracy.

The estimation error of the traditional ILS method fluctuated widely, and when the estimation results were better, the error in the results could be stabilized at the centimeter level, but most of the test points had an error of more than 1 m. The absolute error distribution of the test points was not uniform, the median of some of the data was close to the upper quartile, and the estimation results for the different calendar elements varied widely. This can lead to the lack of a reference value for the normalized results.

The estimation error value using the TSLS method was stable but greater than that using the proposed method. When the estimation results were good, the error in the results could be stabilized at the centimeter level, but most of the test points reached the meter level and the error was unstable, which can lead to difficulties in estimating the result error size in application scenarios.

To compare the performance of the proposed method and ILS in estimating test point positions, the mean square error (MSE) was used as an evaluation indicator: (15)$$MSE=\;\;E\overset{m}{\underset{j=1}{\sum }}\left[{\left(\hat{x_{j}}-\tilde{x}\right)}^{2}+{\left(\hat{y_{j}}-\tilde{y}\right)}^{2}+{\left(\hat{z_{j}}-\tilde{z}\right)}^{2}\right]\;\;\;\;\;\;\;\;j=\;\;1,2,\ldots 100.$$
where $\hat{z_{j}}\;\mathrm{and}\;\tilde{z}$ are $\hat{h_{j}}\;\mathrm{and}\;\tilde{h},$ respectively. $\left(\hat{x_{j}},\hat{y_{j}},\hat{h_{j}}\right)$ represent the test point estimate coordinates in the first j epochs, $\left(\tilde{x},\tilde{y},\tilde{h}\right)\;$ represent the true coordinates, and j indicates the different observation epochs. The MSE is effective in reflecting position accuracy statistics. The MSE for each test point is listed in Table 3. Compared with the traditional ILS and TSLS methods, the proposed method resulted in a mean improvement percentage for the MSE of 63.54%, and all estimation improvements were above 30%. It can be seen that the accuracy and stability of the proposed method were better than those of the ILS method. The TSLS method itself has a large MSE, which may be due to the fact that the calculation results tend to diverge without control. The improvement in the MSE was very obvious after the new method ensured the convergence of the results.

### 4.2. Simulation and Estimation of Different Ranging Error Positions

The position estimate simulation was based on observations generated from a determined value for the error variance. To verify the proposed method position results according to different distance variance values, we conducted a group of distance measurements of a fixed test point and calculated its position using the proposed method. Considering that the simulation data of the last experiment were generated with a determined value of 25 cm^2^ error variance, a simulation using data with different variances was necessary to verify the proposed method’s ability to calculate data with different distance variances. In the simulation, a group of distance measurements of a fixed test point were generated, and the position was calculated from the measurements using the proposed method. The test point in the middle of the measurement space was selected: $P\left(7,7,1.5\right)$. The coplanar base-station configuration was the same as that shown in Figure 2, and the range variance changed from σ = 0 cm to σ = 10 cm. One hundred epoch observation values were generated for each variance. For each epoch observation, a position estimate and the MSE were calculated. The position results using the three methods with different ranging errors are compared in Figure 5. The subplot titles indicate the ranging errors of the corresponding simulation experiments, and the two sets of data were the mean and variance values. In each case, the localization error and the variance of the proposed method were better than those other two methods.

To present the MSE of the position results using the proposed method with different ranging errors in detail, the change curve for the MSE with the ranging variance is shown in Figure 6, The line between the points indicates the trend of change.

The MSE increased linearly as the ranging error increased. The MSE was less than 0.13 when the range error was in the centimeter range, and this accuracy is sufficient for general positioning scenarios. For higher accuracy requirements, the corresponding range accuracy matching is required.

### 4.3. Dynamic Position Experimental Campaign

A DW1000 UWB system was selected as the base station for ranging. Four base stations were deployed in the test space, and one unknown mobile station was moved around the test space. Base stations were deployed at the four corners of a square indoor scene (12 m × 12 m), and the height of the base stations was approximately 2 m, as shown in Figure 7a. The coordinates were as follows: A1(1, 1, 2.066), A2(1, 13, 2.081), A3(13, 13, 2.013), and A4(13, 1, 1.997). The unknown station was assumed to be on a mobile stand with a height of approximately 1.5 m, and it was connected to a computer to record positioning data, as shown in Figure 7b. The test space was in a building with a patio structure. The ceiling was very high, which helped to avoid ranging errors caused by signal reflection from the ceiling. The floor of the test space was square, and the base stations were laid in the four corners of the open space with stands at a height of about 2 m. The unknown points moved below the surface that the base stations lay on. A top view of the layout of the base stations and the moving trajectory of the unknown point is shown in Figure 8.

In the experiment, the experimenter moved a ground marker around the experimental site and collected real-time UWB data. The trajectory path was almost a circle, as shown in Figure 8. The measurement frequency was 10 Hz, and a total of 141 ephemeris distance measurement results were recorded. One positioning result was calculated for each ephemeris, and all results were matched with the sampling time to generate the mobile station moving track.

The positioning trajectories calculated with the proposed method, the ILS method, and the TSLS method are shown in Figure 9. During the measurement, the accuracy of the positioning results for all ephemeris elements was counted, and the average errors, maximum errors, and minimum errors of the three methods are compared in Table 4. In agreement with the simulation results, the positioning accuracy of the proposed method was significantly higher than that of the ILS and TSLS methods, with an error of less than 5.5 cm and an average error of approximately 2.4 cm, which is suitable for the needs of practical applications. The ILS results deviated greatly from the real measurements, with an average error of over 1 m. The TSLS results converged to the height around the base station, which was caused by the multipolar value of the objective function.

## 5. Performance Analysis

The base-station configuration affects the objective function of the location estimation, and this effect is quantitatively analyzed in this section. Firstly, a theoretical derivation of the upper limit of the localization accuracy of the least-squares criterion, which is closely related to the base-station configuration, is presented. Secondly, the reasons why the iterations have difficulty in converging correctly during the iterative solution process for the positioning equations when the base stations are coplanar are described. In addition, this section provides describes an experimental comparison of the position estimation results in different base-station deployment scenarios.

### 5.1. Least-Squares Algorithm Positioning Accuracy Analysis

In the location estimation process, the coordinates of the unknown point are assumed to be $X={\left[x,y,h\right]}^{T}$. Based on the known base-station coordinates and the distance observations, the unknown parameters, observations, and base-station coordinates are jointly listed in the following equation: $\hat{d_{i}}=f_{i}\left(X\right)$. The Taylor expansion of the part to the right of the equals sign with the approximate estimated value $X_{o}={\left[x_{o},y_{o},h_{o}\right]}^{T}$ is obtained as a system of equations for the observations: (16)$$\left[\begin{array}{c} {\hat{d}}_{1}\\ \vdots \\ {\hat{d}}_{n} \end{array}\right]=\left[\begin{array}{ccc} -\frac{\left(x_{1}-x\right)}{\sqrt{d_{1o}^{2}}} & -\frac{\left(y_{1}-y\right)}{\sqrt{d_{1o}^{2}}} & -\frac{\left(h_{1}-h\right)}{\sqrt{d_{1o}^{2}}}\\ \vdots & \vdots & \vdots \\ -\frac{\left(x_{n}-x\right)}{\sqrt{d_{no}^{2}}} & -\frac{\left(y_{n}-y\right)}{\sqrt{d_{no}^{2}}} & -\frac{\left(h_{n}-h\right)}{\sqrt{d_{no}^{2}}} \end{array}\right]\left[\begin{array}{c} x\\ y\\ h \end{array}\right]-\left[\begin{array}{ccc} -\frac{\left(x_{1}-x\right)}{\sqrt{d_{1o}^{2}}} & -\frac{\left(y_{1}-y\right)}{\sqrt{d_{1o}^{2}}} & -\frac{\left(h_{1}-h\right)}{\sqrt{d_{1o}^{2}}}\\ \vdots & \vdots & \vdots \\ -\frac{\left(x_{n}-x\right)}{\sqrt{d_{no}^{2}}} & -\frac{\left(y_{n}-y\right)}{\sqrt{d_{no}^{2}}} & -\frac{\left(h_{n}-h\right)}{\sqrt{d_{no}^{2}}} \end{array}\right]\left[\begin{array}{c} x_{o}\\ y_{o}\\ h_{o} \end{array}\right]+\left[\begin{array}{c} \sqrt{d_{1o}^{2}}\\ \vdots \\ \sqrt{d_{no}^{2}} \end{array}\right]$$

In Equation (13), we can note that the unknown ${\left[x,y,h\right]}^{T}$ is X, $\hat{L}$ is the matrix representation of the vector ${\left[\hat{d_{1}},\ldots ,\hat{d_{n}}\right]}^{T}$, and A is the matrix of X coefficients. The constant term matrix is denoted C. The equation is expressed in matrix form as $\hat{L}=AX+C$, which gives the error equation: $V=AX+C-\hat{L}$. The expression for X is obtained as follows: (17)$$X={\left(A^{T}A\right)}^{-1}A^{T}\left(\hat{L}-C\right)$$

Q denotes the covariance and the subscript denotes the covariance object. According to the covariance propagation law, the covariance of X is $Q_{\Delta X\Delta X}$   $={\left(A^{T}A\right)}^{-1}A^{T}Q_{\hat{L\hat{L}}}A{\left({\left(A^{T}A\right)}^{-1}\right)}^{T}$.

Each observation is assumed to have the same precision and variance of 1, which means that $Q_{\hat{L\hat{L}}}=\frac{1}{\sigma_{0}^{2}}D_{\hat{L\hat{L}}}=I$. The covariance array of the solution is: (18)$$Q_{\Delta X\Delta X}={\left(A^{T}A\right)}^{-1}A^{T}IA{\left({\left(A^{T}A\right)}^{-1}\right)}^{T}={\left(A^{T}A\right)}^{-1}=\left[\begin{array}{ccc} \sigma_{x}^{2} & \sigma_{xy} & \sigma_{xh}\\ \sigma_{yx} & \sigma_{y}^{2} & \sigma_{yh}\\ \sigma_{zx} & \sigma_{zy} & \sigma_{h}^{2} \end{array}\right]={\left[\begin{array}{ccc} a_{1} & b_{1} & c_{1}\\ a_{2} & b_{2} & c_{2}\\ a_{3} & b_{3} & c_{3} \end{array}\right]}^{-1}$$

Assuming that multiple base stations are deployed coplanarly in the horizontal direction (that is, $h_{1}=h_{i}=h_{n}$), consider the expression for the accuracy of the h-positioning results: (19)$$\sigma_{h}^{2}=\frac{a_{1}b_{2}-a_{2}b_{1}}{\left(a_{2}b_{3}-a_{3}b_{2}\right)c_{1}+\left(a_{3}b_{1}-a_{1}b_{3}\right)c_{2}+\left(a_{1}b_{2}-a_{2}b_{1}\right)c_{3}}$$

This includes the variance and mutual covariance of the unknown three-parameter solution, as expressed by: $$\begin{array}{c} a_{1}=\sum_{i=1}^{n}\frac{{\left(x_{i}-x_{o}\right)}^{2}}{d_{io}^{2}},b_{2}=\sum_{i=1}^{n}\frac{{\left(y_{i}-y_{o}\right)}^{2}}{d_{io}^{2}},c_{3}=\sum_{i=1}^{n}\frac{{\left(h_{i}-h_{o}\right)}^{2}}{d_{io}^{2}},b_{1}=a_{2}=\left(\sum_{i=1}^{n}\frac{\left(x_{i}-x_{o}\right)\left(y_{i}-y_{o}\right)}{d_{io}^{2}}\right),\\ c_{1}=a_{3}=\left(\sum_{i=1}^{n}\frac{\left(x_{i}-x_{o}\right)\left(h_{i}-h_{o}\right)}{d_{io}^{2}}\right),b_{3}=c_{2}=\left(\sum_{i=1}^{n}\frac{\left(h_{i}-h_{o}\right)\left(y_{i}-y_{o}\right)}{d_{io}^{2}}\right). \end{array}$$

As seen from the expressions, the solution on the h-axis is a function of only the base station’s position relative to the point of interest.

The variance can be used to effectively assess the degree of dispersion in the algorithm’s in the results prior to calculation. To a certain extent, it reflects the reliability of the results. Equation (16) shows that the variance in the calculated results increases as the base station approaches coplanarity, and the localization accuracy decreases to the point where the correctness of the results cannot be determined. This is precisely the reason why the localization results vary when facing different base-station configurations using the ILS method.

### 5.2. Impact of the Iterative Method on Base-Station Coplanarity

Using a specific calculation process, this section analyses the localization solution for the base station.

When the base stations are coplanar, the objective function is nonconvex, the objective function solution constantly converges to the point where the first-order derivative is zero, and the extreme value is not unique, leading to difficulties in finding the iterative solution. If we set up multiple base stations at the same height $h_{1}=\cdots =h_{n}$, the first-order derivative of the h direction is expressed as follows: (20)$$\frac{\partial V^{T}V}{\partial h}=2\overset{n}{\underset{i=1}{\sum }}\left(h-h_{i}\right)\left(1-\frac{\hat{d_{i}}}{\sqrt{{\left(x-x_{i}\right)}^{2}+{\left(y-y_{i}\right)}^{2}+{\left(h-h_{i}\right)}^{2}}}\right)$$

Assuming there are four reference stations laid out on a common surface, one test point will be located at the middle of the test space. The values of fixed *x* and *y* are consistent with the true coordinates, and the first-order-derivative changes in the h direction are plotted in Figure 10. The *x*-axis in the figure indicates the *h* variation and the *y*-axis indicates the magnitude of the first-order derivative.

As shown in Figure 10, the first-order partial derivatives appear numerically as multiple zeros, $h=h_{ANC}$, $h=\tilde{h_{TAG}}$, and $h=2h_{ANC}-\tilde{h_{TAG}}$. A previous study [25] explored the multipolarity case can affect the objective function in the case of base-station coplanarity, where the iterative process may fall into the vicinity of different poles due to the presence of multiple poles.

The selection of the initial iteration value is one of the important factors that affect the iterative process. With the initial value selection method proposed in this paper, it is easier to obtain the correct solution by first selecting a point near the correct extremum point in the coplanar direction in the calculation. Controlling the step size of the iterations helps the iterative results to converge. The iterative direction control is achieved through the combination of the limitation of the iterative initial value range and the determination of the first-order derivatives of the computational process. With the combination of these two aspects, the correctness of the computational results is guaranteed.

The first-order derivative of the objective function has zeros $h=2h_{ANC}-\tilde{h_{TAG}}$ around the true value; in addition, when the base stations are completely coplanar, $h_{ANC1}=\cdots =h_{ANCn}$, the first-order derivative increases the zero $h=h_{ANC}$ point number in the localization space. The maximum value of the mutual difference in the base-station heights is used as the representation of the degree of coplanarity, and a value closer to 0 indicates that the base station is deployed in a nearly coplanar fashion. As the degree of coplanarity gradually increases from 0, the first-order-derivative value close to the coplanarity elevation value also gradually increases, and the iteration cannot take a minimal value close to zero. When not considering the ranging error, different iterative extrema can be judged according to the number of iterative zeros that are close to zero, and the extrema that converge to the correct solution can be selected after discrimination. Based on the iterative threshold setting and considering the distance error, almost-coplanar base-station deployment can be achieved within a particular activity space. After calculation and deduction, the base-station elevation can be determined by: (21)$$h-h_{ANC}\left(\frac{\sigma_{d}d_{r}}{\sigma_{dr}d_{max}}\right)\leq h_{ANC}\leq h+h_{ANC}\left(\frac{\sigma_{d}d_{r}}{\sigma_{dr}d_{max}}\right)$$
where $\sigma_{d}$ is the relative maximum error value of the distance observation, $\sigma_{dr}$ is the relative error value of the current distance observation, $d_{r}$ is the current distance observation value, $d_{max}$ the maximum distance observation value of the observation space, $h_{ANC}\;$ is the base-station height, and h is the average height of the base station.

When the base-station height is active in the above range, the first-order derivative of the objective function is numerically difficult to distinguish from the extreme values close to the base-station height and the near-real values. Therefore, it is difficult to determine the position result. Multiple extreme points can be clearly distinguished when the base station moves beyond this range. The range is expressed in relation to the target movement point and the height at which the base station is deployed in the measurement environment.

In the above theoretical analysis, the error in the measurements caused by the distance measuring equipment was not considered, and the first-order derivative can be set to 0 under the ideal distance observation condition. However, due to the limited observation accuracy of instruments, there will be errors in the actual measurement results. The observation results can also be disturbed by the environment so that the mean value of the observation error is not zero, resulting in the first-order derivative not being set to zero. In the iterative calculation process, the iterative stopping threshold needs to take into account the above error factors to ensure that the position results can be obtained. The threshold value of the algorithm can also be applied in positioning environments where the base stations are not strictly coplanar and can change within a certain range near the deployment plane.

### 5.3. Simulation Results for Different Base Station Layouts

In this section, the influence of the base-station configuration on the positioning accuracy is presented.

Four base stations were set in two scenarios. First, all base stations were set at a level of 6 m, and the base station coordinates were (1,1,6), (13,13,6), (1,13,6), and (13,1,6). Second, four base stations were evenly distributed on two elevation planes; specifically, (1,1,0), (1,13,6), (13,13,0), and (13,1,6). The spatial midpoint (7,7,4) was selected as the test point in the two scenarios, and two different scenario configurations are shown in Figure 11.

The ILS algorithm and fixed Newton algorithm were used to estimate the test point position. The estimated position result was obtained over 1000 observed epochs. The result showed that, in both algorithms, the positioning accuracy in a coplanar layout was lower than that in a uniform layout.

The estimation observation error conformed to a Gaussian distribution with a mean of zero $\Delta d\sim N\left(0,\sigma^{2}\right)$. For real-time observations, the distance error variance is related to the observation equipment and electromagnetic environment. In most line-of-sight observations, the observation mean error is zero. This is the reason why we set μ=0 and $\sigma^{2}=0.1\mathrm{cm}$ in our experiment.

The simulation was used to determine the position accuracy in different scenarios. As shown in Figure 12, the accuracy of the estimated test point position in the coplanar scenario was significantly lower than that in the uniform scenario, especially using the iterative least-squares algorithm. In the coplanar scenario, the model was unable to converge to the correct solution.

With regard to the positioning error of the two algorithms, as we expected, the proposed method performed better than the traditional ILS method. Positioning errors with the same CDF were compared. The proposed method outperformed the traditional method in both scenarios. In the uniform scenario, all estimated test point positioning errors were less than 20 cm for both methods. In coplanar scenarios, the proposed method had an error of less than 20 cm for 100% of the results as compared to approximately 70% for the traditional ILS algorithm. Similarly to the mathematical formula in Equation (16) above, the fixed point in the coplanar base-station scenario could not be obtained with the traditional ILS algorithm.

The testing of the two methods was described in Section 4. Simulations were carried out for random points with different spatial heights, and experimental tests were carried out for a fixed-height stand-erected mobile station. The results were consistent with the discussion in this section, and the positioning errors are presented in Table 5. The new method was able to exercise control in the elevation direction and improve the accuracy of the positioning results.

## 6. Conclusions

In the indoor point-solution problem, base-station deployment affects the calculation results. In a nearly coplanar base-station deployment environment, the traditional algorithm is not applicable. In order to resolve positioning errors in such an environment, this paper mainly focused on the following aspects:
(1)Based on the observation equation, the influence of the initial value on the convergence result during calculation was analyzed, and a method for selecting the initial value under a coplanar base-station condition was proposed to facilitate the correct convergence of the iterative results;(2)Considering the observation conditions and positioning accuracy requirements, this paper proposed an iterative convergence control method. The iterative step length was adjusted to avoid iterative scattering; the intermediate value of iteration was determined to control the direction of iteration and ensure that the result converged to the correct solution;(3)The mathematical derivation of the localization accuracy of the least-squares criterion settlement method was carried out. Experiments were conducted for different base-station deployment scenarios. The results showed that the new method improved the convergence performance by about 15% in the uniform scenario and about 30% in the coplanar base-station scenario.

The newly proposed method was tested in both a simulation and experiment. The average positioning accuracy of the simulation results was 7.5 cm compared to 25.3 cm for the original method, and the positioning accuracy of the new method was improved by about 63.54%. In the actual positioning experiment, the average error of the new method was 0.024 m, while the average error of the traditional method was 1.35 m.

The results show that the newly proposed method can effectively improve the positioning accuracy when facing the positioning problem that arises from the near-coplanar placement of base stations.

## Figures and Tables

**Figure 1 sensors-22-09634-f001:**
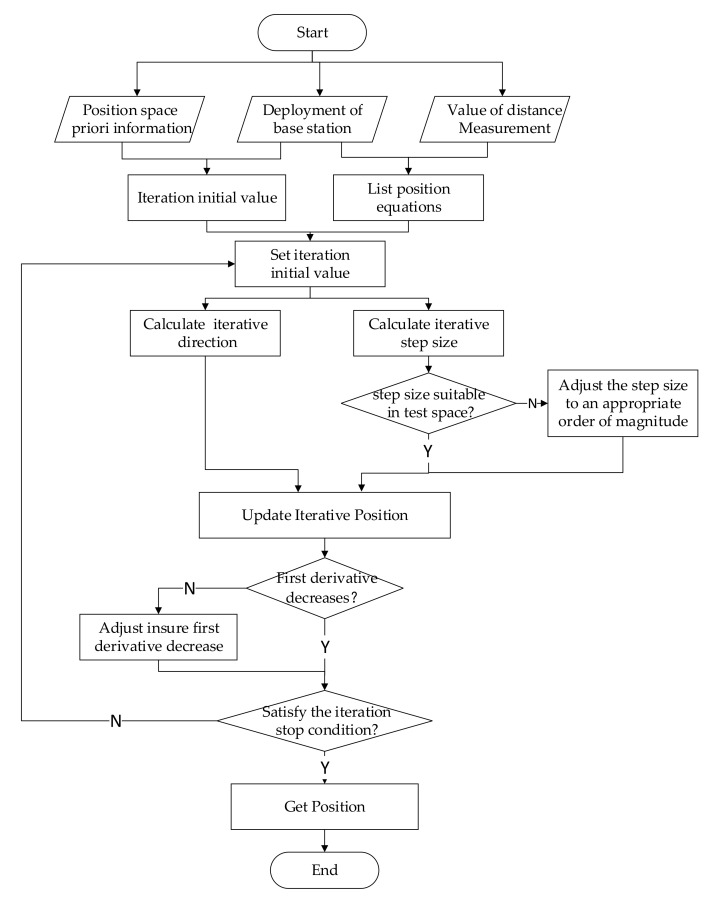
New algorithm flow chart.

**Figure 2 sensors-22-09634-f002:**
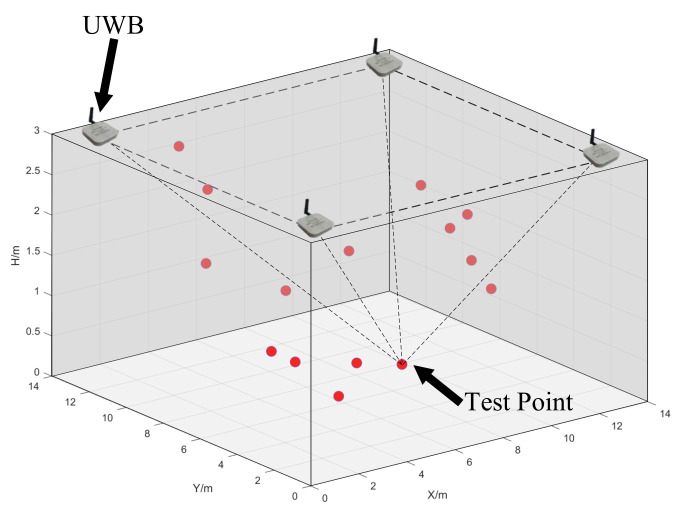
Configuration with four base stations and fifteen test points.

**Figure 3 sensors-22-09634-f003:**
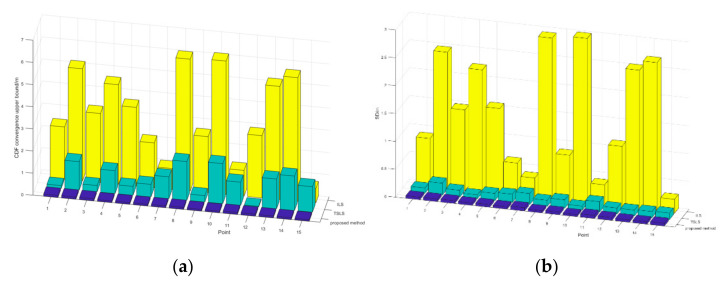
Statistical comparison of the three methods. (**a**) the comparison of CDF coverage upper bound (**b**) the comparison of SD.

**Figure 4 sensors-22-09634-f004:**
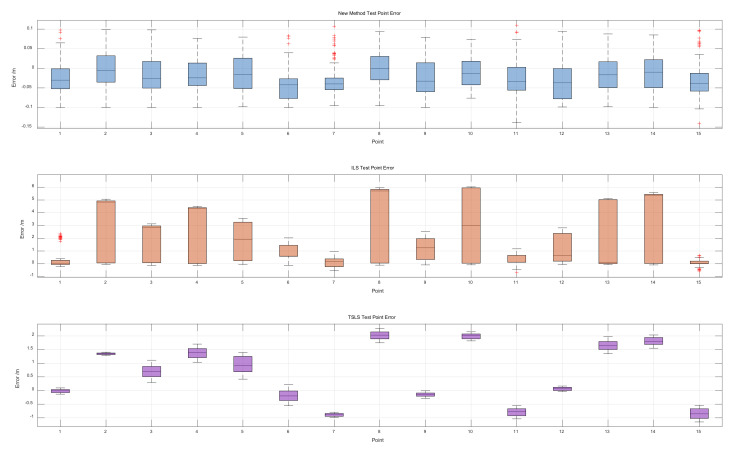
Estimated position errors for the three methods.

**Figure 5 sensors-22-09634-f005:**
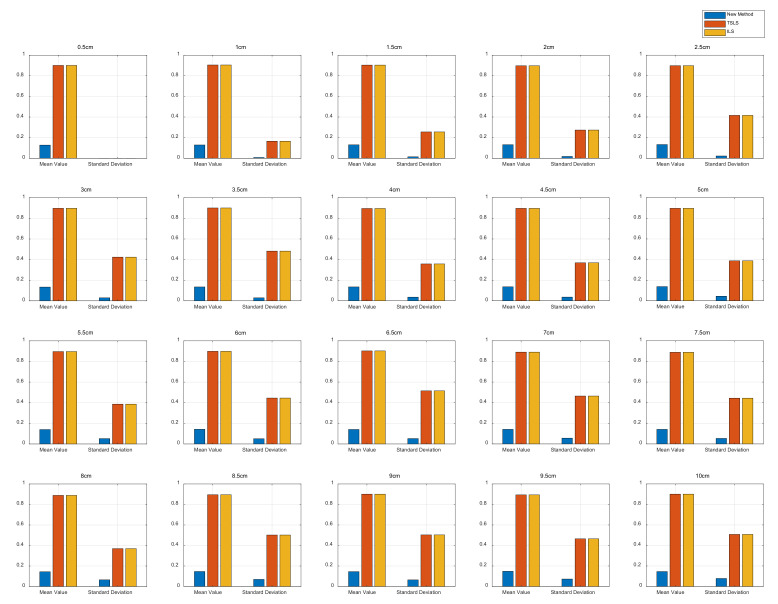
Comparison of position results with different ranging errors.

**Figure 6 sensors-22-09634-f006:**
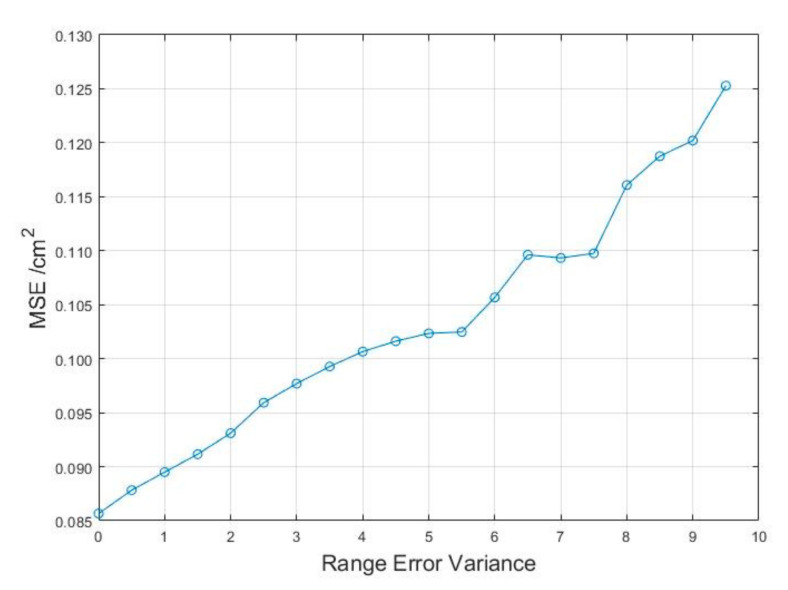
MSE change with ranging variance for the proposed method results.

**Figure 7 sensors-22-09634-f007:**
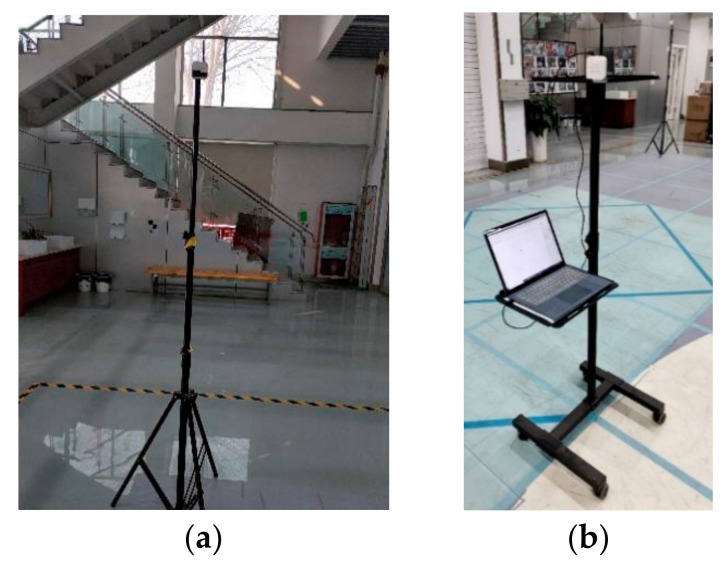
Erection of base stations and mobile station. (**a**) Base station (**b**) unknown station.

**Figure 8 sensors-22-09634-f008:**
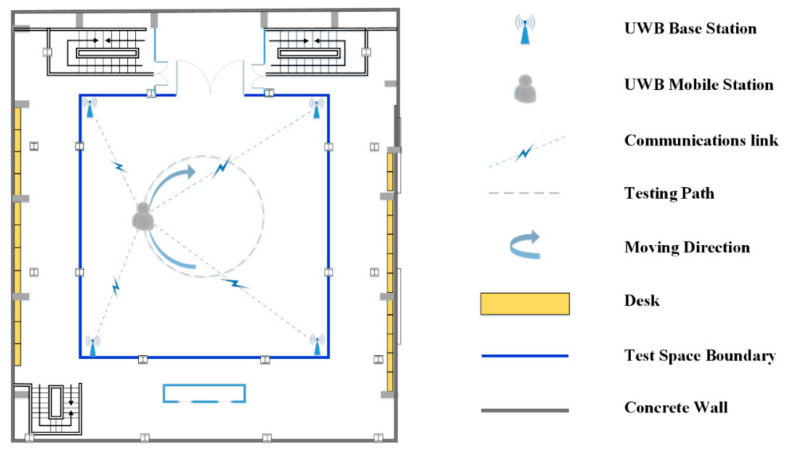
Test environment top-view diagram.

**Figure 9 sensors-22-09634-f009:**
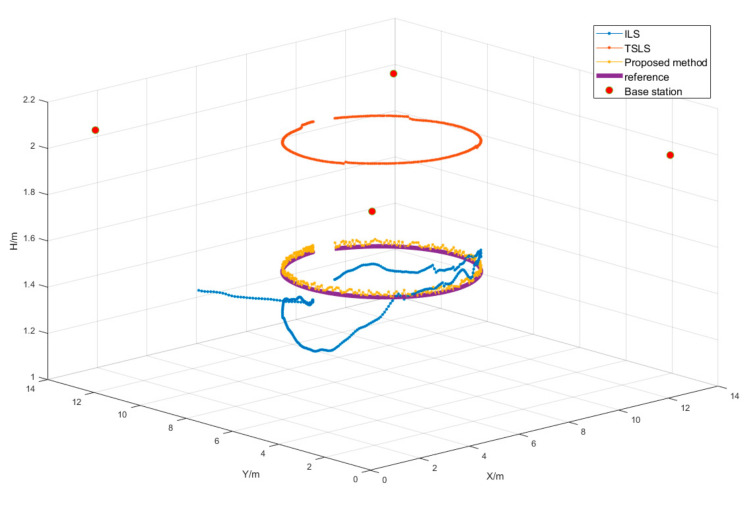
Measured results diagram.

**Figure 10 sensors-22-09634-f010:**
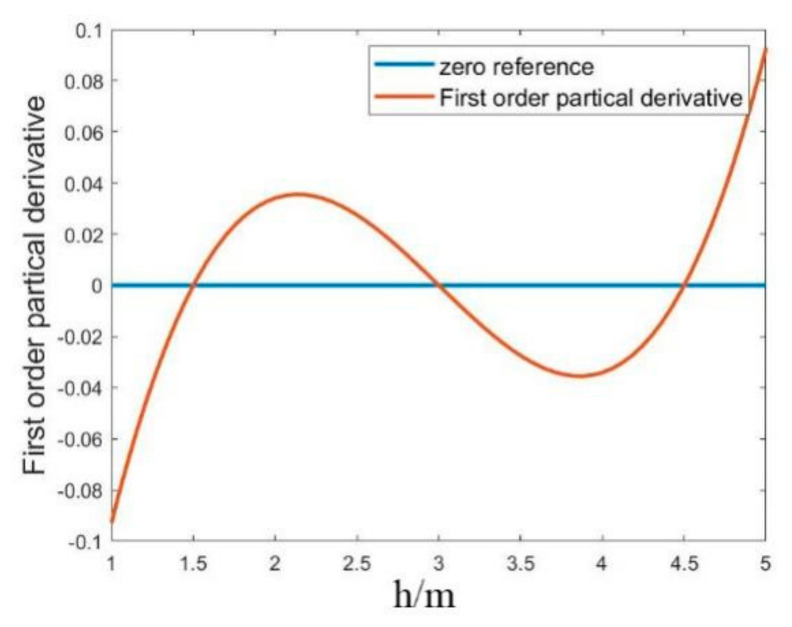
First-order derivative of the objective function in the coplanar direction.

**Figure 11 sensors-22-09634-f011:**
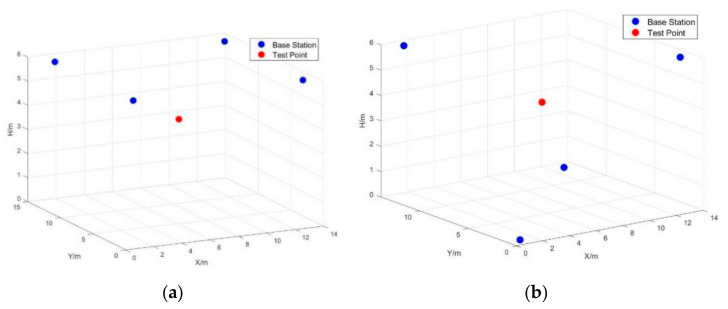
Different scenario configurations. (**a**) coplanar scenario (**b**) uniform scenario.

**Figure 12 sensors-22-09634-f012:**
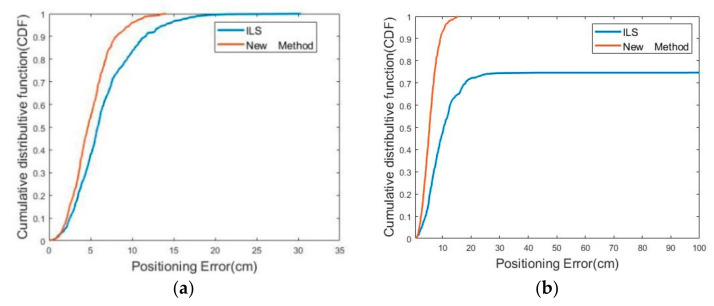
CDFs for two methods. (**a**) Uniform scenario (**b**) Coplanar scenario.

**Table 1 sensors-22-09634-t001:** Multi-epoch results statistics.

Test Point	Maximum Error in Position/m			Standard Deviation/m		
	Proposed Method	TSLS	ILS	Proposed Method	TSLS	ILS
1	0.098	0.129	2.364	0.045	0.088	0.868
2	0.099	1.285	5.066	0.048	0.051	2.449
3	0.098	0.286	3.124	0.049	0.072	1.448
4	0.076	1.034	4.501	0.043	0.052	2.200
5	0.08	0.400	3.553	0.048	0.058	1.541
6	0.083	0.566	2.025	0.044	0.064	0.596
7	0.083	0.983	0.947	0.06	0.215	0.362
8	0.093	1.749	5.96	0.039	0.045	2.905
9	0.079	0.289	2.538	0.047	0.092	0.837
10	0.074	1.808	6.026	0.036	0.037	2.967
11	0.093	1.048	1.169	0.058	0.202	0.375
12	0.094	0.039	2.812	0.052	0.064	1.098
13	0.088	1.343	5.112	0.043	0.045	2.499
14	0.085	1.553	5.582	0.045	0.051	2.668
15	0.073	1.150	0.646	0.052	0.154	0.250

**Table 2 sensors-22-09634-t002:** Three methods for estimating the position results.

Test Point	Reference	ILS δz/m	TSLS δz/m	Proposed Method δz/m
1	(1.248, 7.284, 1.96)	0.466	−0.132	−0.050
2	(6.038, 5.374, 0.553)	2.543	1.281	−0.003
3	(7.783, 8.064, 1.551)	1.683	0.282	−0.025
4	(7.308, 10.866, 0.821)	2.291	1.032	−0.021
5	(9.722, 2.896, 1.44)	1.763	0.397	−0.011
6	(8.932, 3.157, 2.396)	0.908	−0.570	−0.068
7	(1.621, 7.683, 2.811)	0.119	−0.986	−0.077
8	(5.778, 6.012, 0.097)	3.047	1.747	−0.001
9	(7.579, 1.174, 2.12)	1.204	−0.292	−0.049
10	(9.638, 7.617, 0.048)	2.993	1.806	−0.012
11	(7.165, 3.367, 2.869)	0.329	−1.052	−0.049
12	(10.194, 5.744, 1.879)	1.272	−0.041	−0.039
13	(4.781, 8.351, 0.501)	2.224	1.340	−0.016
14	(5.774, 8.356, 0.295)	3.266	1.549	−0.010
15	(2.698, 10.638, 2.979)	0.059	−1.153	−0.105

**Table 3 sensors-22-09634-t003:** Comparison of the MSEs of the three methods.

Test Point	ILS MSE/m	TSLS MSE/m	Proposed Method MSE/m	Improvement over ILS	Improvement over TSLS
1	0.088	0.868	0.001	36.29%	99.89%
2	0.051	2.449	0.002	76.68%	99.91%
3	0.072	1.448	0.001	76.71%	99.92%
4	0.052	2.200	0.001	40.43%	99.97%
5	0.058	1.541	0.001	78.30%	99.91%
6	0.064	0.596	0.002	77.26%	99.74%
7	0.215	0.362	0.002	82.64%	99.45%
8	0.045	2.905	0.001	29.47%	99.96%
9	0.092	0.837	0.002	71.12%	99.82%
10	0.037	2.967	0.001	88.55%	99.97%
11	0.202	0.375	0.002	64.79%	99.50%
12	0.064	1.098	0.001	65.76%	99.90%
13	0.045	2.499	0.001	77.05%	99.96%
14	0.051	2.668	0.001	34.95%	99.96%
15	0.154	0.250	0.001	80.05%	99.52%

**Table 4 sensors-22-09634-t004:** Comparison of the three methods.

Method	Mean Error/m	Max Error/m	Min Error/m
ILS	0.354	0.638	0.228
TSLS	0.565	0.579	0.553
Proposed method	0.024	0.054	0.002

**Table 5 sensors-22-09634-t005:** Comparison of the two methods.

		Mean Error/m	Max Error/m	Min Error/m
**Multi-Point Simulation**	**Proposed Method**	0.0759	0.123	0.018
	**ILS**	0.253	0.456	0.084
**Single-Height Experiment**	**Proposed Method**	0.024	0.054	0.002
	**ILS**	1.354	4.638	0.228

## Data Availability

The data used in this paper were obtained through measurements by the authors. No publicly available data are cited, and data sharing is not applicable to this article.

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
