# Peer review of "An Indoor UWB 3D Positioning Method for Coplanar Base Stations"

_sensors, 2022, doi:10.3390/s22249634_

Round 1
Reviewer 1 Report (New Reviewer)
Authors should do some enhancement to the paper presentation as follows:
1) Section 2 should explain the most related works which can be compare with in the implementation section
2) The related works should be added in section 2. Consequently, the references should be added with most related works.
3) Authors should compare at least with two/three other traditional methods to prove their claim
4) The results section should be reorder and summarized.
5) References are not sufficient.
Author Response
Thank you so much for the valuable advice. Detailed reply please see the attachment.

Reviewer 2 Report (New Reviewer)
The authors propose an iterative method to achieve 3D indoor positioning method considering coplanar 2 base stations. The authors should improve the manuscript based on the following comments.
1. The authors should highlight their contributions in the part of introduction.
2. How about the convergence speed of the iterative method? Is it suitable for tracking a fast moving target?
3. The authors should compare their approaches to some SOTA methods based on UWB, such LS, Non-LS, EKF, etc.
4. How does NLOS affect the method?
Author Response
Thank you so much for your careful review, Please see attachment for detailed response.

Reviewer 3 Report (New Reviewer)
The authors propose a novel iterative method to estimate the position under the coplanar base station scenarios. The improvement is shown that the test point accuracy is 25 improved by an average of 63.54% as compared to the conventional positioning algorithm when the base stations are coplanar. Hence, it can be accepted after revision.
Comment:
1. Section 2 and 3 introduce the method, but some references are necessary and need to be added.
2. Keep the font type consistent in Figure 1.
3. Figure 3 and 5 are so fuzzy and needs to be revised.
4. More information can be added to the Table 2 for the better comparision.
Author Response
Thank you so much for your careful review, Please see the attachment for the detailed response.

Round 2
Reviewer 2 Report (New Reviewer)
The authors have addressed my comments. Well done.
This manuscript is a resubmission of an earlier submission. The following is a list of the peer review reports and author responses from that submission.
Round 1
Reviewer 1 Report
This paper proposes a novel iterative method to estimate the position in coplanar base station deployment scenarios. The proposed method works for underground scenarios.
It is suggested that “Intro” section is split in two sections – the second one will be Background where all this info about the related work can be included (from the second paragraph until “According to the above mentioned problems”) The authors should rewrite the intro holding only the research motivation (why it is important to do this research), a small paragraph highlighting the existing research and which research gap they will fill, How they have worked (a summary of their methodology e.g. the experiment in never mentioned), a summary of the contribution (the paper lacks the practical implications – who and how they will use this research) and then the paper outline.
all this info about the existing research can be included (from the second paragraph in intro until “According to the above-mentioned problems”) in Background section together with section 2. The traditional method. It is also suggested that the authors will be more specific about works concerning underground scenarios. Haven’t they found any such works?
Important info is missing in the experiment: when a test deployment is conducted, the deployment case should be described. Please give a brief description of your case study, how was this place where you placed the base stations
After the performance analysis, I would include a summary of all the Results that have been included in several points in the text. Perhaps a table could help.
In the conclusion section, a discussion of the results is missing. How are the results compared with results in related papers?
Author Response
Thank you for your comments, please see the attached reply.

Reviewer 2 Report
See attached file

Author Response
谢谢您的意见。您的回复见附件。

Round 2
Reviewer 2 Report
The revised version only containes minor changes which did not address the problems with the original version. The ILS method is known to perform poorly, so it is not appropriate as a reference for performance estimation. While the proposed method is much improved over ILS, the standard linearised least-squares method generally performs better for the examples in this paper. For example, the 15 point case study, the LLS gave better performance in 10 cases, similar in 3, and worse in only 3 cases.
The paper does not discuss VDOP, which is the standard method of quatifying positional accuracy with Gaussian ranging errors. This shows that the accuracy is poor when the Tag z-coordinate approaches the z-coordinate of the base station plane. No "mathematical magic" can overcome this behaviour, but can only minimise the effect, as in this paper.
The experiments with the UWB equipment have little relevance to the main thrust of the paper, namely the effect on accuracy as the z-coordinate approaches the base station z-coordinate. Further, the measurements, simulations and theory are in near ideal conditions, and ignores multipath and other effects such as equipment measurement errors such as ranging bias. Such circumstances are very important in the suggested environment in underground mines.
While the suggested method does improve the accuracy of the ILS method, the overall thrust of paper is flawed in that it cannot significantly improve on existing methods, and indeed is often worse.